# Stem rust resistance in wheat is suppressed by a subunit of the mediator complex

Colin W. Hiebert[1,11✉], Matthew J. Moscou [2,11✉], Tim Hewitt[3,4], Burkhard Steuernagel[5], Inma Hernández-Pinzón[2], Phon Green[2], Vincent Pujol[6], Peng Zhang [3], Matthew N. Rouse[7], Yue Jin [7], Robert A. McIntosh[3], Narayana Upadhyaya [4], Jianping Zhang [4], Sridhar Bhavani[8], Jan Vrána[9], Miroslava Karafiátová [9], Li Huang[10], Tom Fetch[1], Jaroslav Doležel [9], Brande B.H. Wulff [5], Evans Lagudah [4✉] & Wolfgang Spielmeyer[4✉]

Stem rust is an important disease of wheat that can be controlled using resistance genes. The gene *SuSr-D1* identified in cultivar 'Canthatch' suppresses stem rust resistance. *SuSr-D1* mutants are resistant to several races of stem rust that are virulent on wild-type plants. Here we identify *SuSr-D1* by sequencing flow-sorted chromosomes, mutagenesis, and map-based cloning. The gene encodes Med15, a subunit of the Mediator Complex, a conserved protein complex in eukaryotes that regulates expression of protein-coding genes. Nonsense mutations in Med15b.D result in expression of stem rust resistance. Time-course RNAseq analysis show a significant reduction or complete loss of differential gene expression at 24 h post inoculation in *med15b.D* mutants, suggesting that transcriptional reprogramming at this time point is not required for immunity to stem rust. Suppression is a common phenomenon and this study provides novel insight into suppression of rust resistance in wheat.

[1] Agriculture and Agri-Food Canada, Morden Research and Development Centre, 101 Route 100, Morden, MB R6M 1Y5, Canada. [2] The Sainsbury Laboratory, University of East Anglia, Norwich Research Park, Norwich NR4 7UK, UK. [3] Plant Breeding Institute Cobbitty, University of Sydney, Private Bag 4011, Narellan, NSW 2567, Australia. [4] CSIRO Agriculture & Food, GPO Box 1700, Canberra, ACT 2601, Australia. [5] John Innes Centre, Norwich Research Park, Norwich NR4 7UH, UK. [6] Research School of Biology, The Australian National University, Acton, ACT 2601, Australia. [7] USDA-ARS, Cereal Disease Laboratory, University of Minnesota, St. Paul, MN 55108, USA. [8] CIMMYT, ICRAF House, United Nations Avenue, Gigiri, Village Market, Nairobi 00621, Kenya. [9] Institute of Experimental Botany, Czech Academy of Sciences, Centre of the Region Haná for Biotechnological and Agricultural Research, Šlechtitelů 31, 779 00 Olomouc, Czech Republic. [10] Department of Plant Sciences and Plant Pathology, Montana State University, Bozeman, MT 59717, USA. [11]These authors contributed equally: Colin W. Hiebert, Matthew J. Moscou. ✉email: Colin.Hiebert@AGR.GC.CA; matthew.moscou@tsl.ac.uk; Evans.Lagudah@csiro.au; Wolfgang.Spielmeyer@csiro.au

Wheat (*Triticum aestivum*) is one of three major cereal crops that supply the majority of calories worldwide, with over 600 million tonnes harvested annually[1]. Wheat is an allopolyploid that includes genomes from three different grass species. A series of sequential hybridization events occurred, with the first at ~0.8 Mya between diploid *T. urartu* (A genome) and an unknown diploid species related to *Aegilops speltoides* (B genome) to form the progenitor of wild allotetraploid emmer wheat (*T. turgidum* subsp. *dicoccoides*; A and B genomes), and the second between emmer wheat and diploid goatgrass (*Ae. tauschii*; D genome) approximately 8000 ya to generate the allohexaploid progenitor of modern common wheat (*T. aestivum*; A, B, and D genomes)[2–4]. Importantly, the three genomes of common wheat (A, B, and D) originate from species that evolved independently from one another for 2–3 million years[5].

Crop diseases are a major limitation to food production. Developing cultivars carrying effective disease resistance genes is a sustainable and environmentally responsible approach to disease management. However, pathogens can evolve virulence to resistance genes emphasizing the need to identify and utilize new resistance genes that are found within a crop and related species. Wheat stem rust (caused by *Puccinia graminis* Pers. f. sp. *tritici* Eriksson & Henning; *Pgt*) is a devastating fungal disease of wheat[6] that has re-emerged as a worldwide threat to wheat production with the evolution of highly virulent races of *Pgt* in Africa, including the Ug99 race group[7–9]. Races of *Pgt* are defined by the combination of host resistance genes for which a given *Pgt* isolate displays either virulence or avirulence[10].

Interaction between the genomes in allopolyploid wheat is associated with the reduction or loss (suppression) of resistance to fungal plant pathogens causing stem rust[11–13], leaf rust[11,14,15], stripe rust[16], and powdery mildew[17,18]. One constraint in wheat breeding is the suppression of stem rust resistance (*Sr*) genes when transferred from diploid and tetraploid ancestors to hexaploid wheat[11–15,19–22]. Suppression of stem rust resistance was first reported in 1980 when it was discovered that the loss of the D genome chromosomes activated resistance to several *Pgt* races that were virulent to the hexaploid wheat cultivar "Canthatch" (CTH)[23]. Initially the D-genome chromosomes were removed from CTH leaving behind the tetraploid (A and B genomes) component, called Tetra CTH[24]. This tetraploid was resistant to several races of *Pgt* that were virulent to hexaploid CTH. Reintroduction of the D-genome by crossing Tetra CTH with *Ae. tauschii* to create a synthetic hexaploid (A, B, and D genomes) resulted in a loss of resistance that was the same as CTH[23]. This example is analogous to instances of *Sr* genes from tetraploid wheat being suppressed when transferred to hexaploid wheat[11–13]. Further analysis of CTH nullisomic and ditelosomic stocks and EMS-derived mutants showed that the suppression of stem rust resistance was conditioned by *SuSr-D1* (*Suppressor of Stem rust resistance1*, D-genome), a single dominant gene on wheat chromosome arm 7DL[23,25,26].

We have a limited understanding of the genes underlying suppression or the resistance genes that are being suppressed. Here, we establish, through genetic mapping, sequenced flow-sorted chromosomes, and mutational analysis that *SuSr-D1* encodes *Med15b.D*, a subunit of the conserved Mediator complex that regulates transcription of protein-coding genes in eukaryotic organisms. Our findings reinforce the complexity of wheat, in particular, the interaction of the sub-genomes and its contribution to unique regulatory processes that impact transcription. As suppression of disease resistance is frequent, the isolation of *SuSr-D1* and understanding its mechanistic role will contribute to developing approaches to understand how the sub-genomes of wheat interact, and how this contributes to

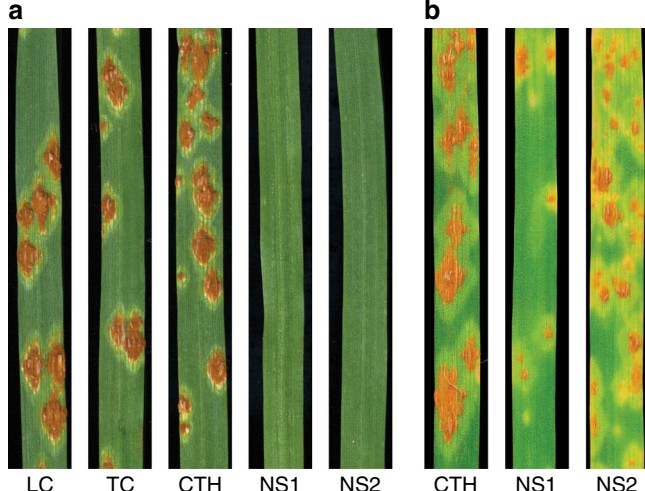

**Fig. 1 Mutation in *SuSr-D1* activates wheat stem rust resistance. a** *Pgt* race QTHJC (C25) showed a qualitative differential response on *susr-d1*-NS1 and *susr-d1*-NS2 compared to wild-type (Canthatch; CTH) and Thatcher (TC) which showed susceptible responses characterized by large uredinia. Little Club (LC) is the susceptible control. **b** *Pgt* race TTKSK (Ug99) shows differential infection responses with large uredinia on CTH and low to intermediate infection types on *susr-d1*-NS1 and *susr-d1*-NS2. All images were taken 14 days post inoculation.

suppressing immunity. Based on this knowledge, future efforts may unlock additional reservoirs of resistance genes for the promotion of sustainable agriculture and food security.

## Results

**Phenotypes of wild-type and mutant stocks.** The key genetic stocks used in these experiments included Canthatch (CTH), Thatcher (TC), CTH mutant lines NS1 and NS2[25], and CTH mutant lines W01, W02, W03, W06, W07, W10, W11, and W12[26], where all ten mutant lines lost *SuSr-D1* activity through EMS mutagenesis. Seedlings of TC, CTH, and NS1 and NS2 showed clear phenotypic differences when inoculated with *Pgt* race QTHJC (C25), with the resistant response of the mutant lines showing no sporulation whereas the wildtype CTH showed a susceptible response with large sporulating pustules at the seedling stage (Fig. 1a). Inoculation of an extended panel of mutants (CTH-W series) with race QTHJC also displayed strong levels of resistance (Supplementary Fig. 1). Inoculating with *Pgt* races from the Ug99 group allowed some rust pustules to develop on mutants, whereas *Pgt* race TRTTF-elicited a susceptible response on CTH, NS1, and NS2 indicating that the resistance expressed in *SuSr-D1* mutant lines was race-specific (Fig. 1b; Supplementary Table 1). Field tests in Kenya on adult plants confirmed seedling test results by showing increased resistance against *Pgt* race TTKSK (Ug99) in all CTH mutants when compared with wild type CTH (Supplementary Table 2).

**Mapping and isolating *SuSr-D1* using enriched DNA sequencing.** Two parallel approaches were used to isolate *SuSr-D1*. In the first approach, chromosome 7D was isolated using flow cytometric sorting from CTH and mutants NS1 and NS2 and sequenced to identify EMS-induced single nucleotide variations (SNVs) (Supplementary Figs. 2–4; Supplementary Table 3)[27,28]. Several SNVs were converted into DNA markers and genetically mapped to identify a minimum physical interval of 1079 kb on chromosome 7D in the Chinese Spring wheat reference sequence that corresponded to the region encompassing *SuSr-D1* (Fig. 2a–c

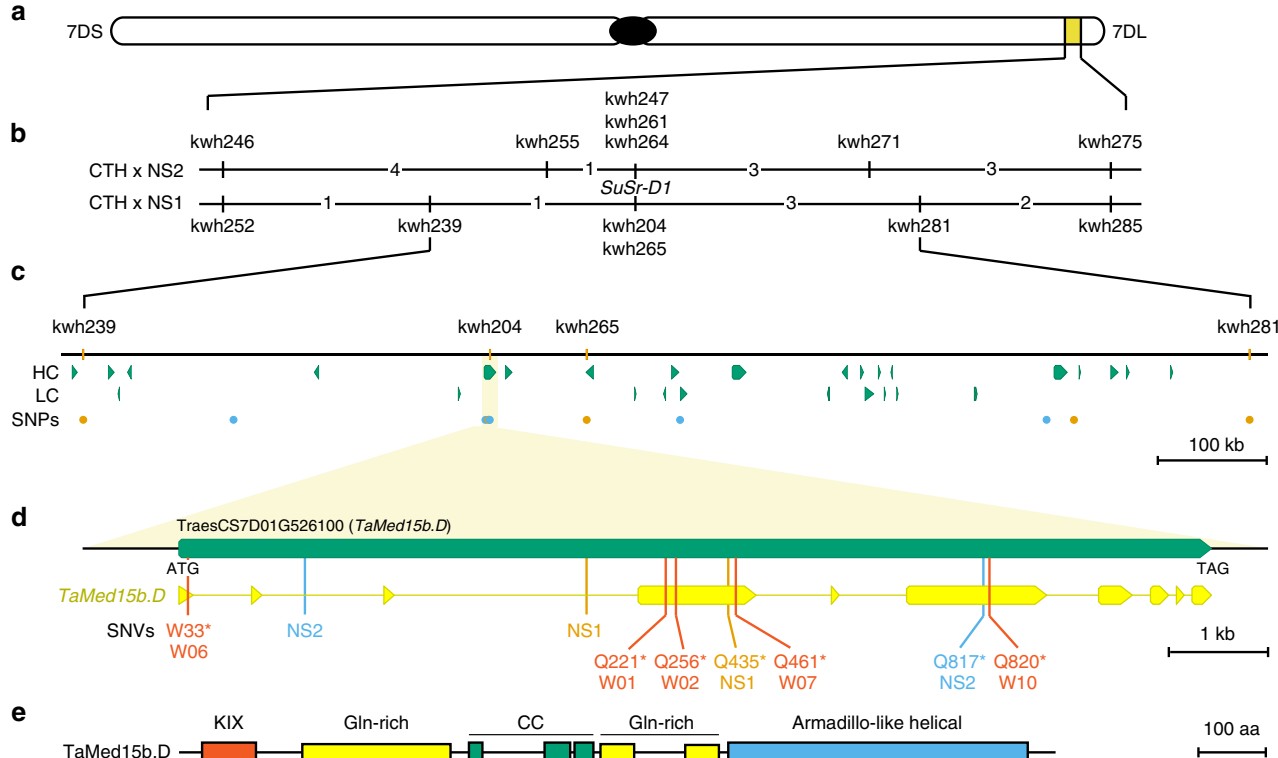

**Fig. 2 SuSr-D1 encodes Med15b.D, a subunit of the mediator complex. a** *SuSr-D1* is located in the distal region of the long arm on chromosome 7D.
**b** Genetic maps of *SuSr-D1* in the NS1 and NS2 DH populations. The values between loci represent the number of recombinants observed in that interval on which the genetic maps in Supplementary Fig. 5 were based. **c** Physical interval encompassing *SuSr-D1* with annotated high (HC) and low (LC) confidence genes (green). **d** Structure of the *Med15b.D* gene encoded by *SuSr-D1*. Exons are represented by thicker segments and introns are represented by thin lines (yellow). NS1 (orange) and NS2 (sky blue) each carry a mutation within an intron as well as the loss-of-function mutations labeled as Q435* NS1 and Q817* NS2. Additional CTH mutations (vermillion) shown are: Q221* W01, Q256* W02, W33* W06, Q461* W07, Q820* W10, and Q820* W11.
**e** *Med15b.D* encodes a 1301 aa protein with KIX (CREB-binding domain[93]; vermillion), Gln-rich (yellow), coiled coil (green), and Armadillo-like helical (blue) domains.

and Supplementary Figs. 5, 6; Supplementary Tables 4–7)[29,30]. Only one gene within the interval carried point mutations in both NS1 and NS2, generating nonsense mutations in *Med15b.D*, a gene encoding *Mediator15*, a subunit of the Mediator complex (Supplementary Fig. 7). In the second approach, de novo assembly of flow-sorted chromosomes from eight mutants identified two mutants with large chromosome deletions and six mutants with nonsense mutations in *Med15b.D* (Fig. 2d, Supplementary Figs. 7, 8). The nonsense mutations in the predicted protein were at several positions from amino acid 33 (W06) to 820 (W10 and W11) (Supplementary Figs. 7, 9). A common mutation in W10 and W11 suggests that the two mutants are sib lines as both possess common SNVs across many contigs[26]. The gene structure of *Med15b.D* spans 10.6 kb with 10 exons that encode a protein of 1301 amino acid residues (Fig. 2d, e). Taken together, independent mutations confirm that inactivation of *Med15b.D* was responsible for the activation of suppressed stem rust resistance.

**Med15 duplication and expression in wheat.** A duplication of *Med15* present on the short (*Med15a*) and long (*Med15b*) arms of group 7 chromosomes in wheat and chromosome 7H in barley (*Hordeum vulgare*) occurred after its divergence from the ancestor of rice (*Oryza sativa*) and *Brachypodium distachyon* (Fig. 3a and Supplementary Fig. 10; Supplementary Table 8). While *B. distachyon* has a single copy of *Med15*, rice has two copies[31]. We identified six *Med15* wheat homologs encompassing two families, *Med15a* and *Med15b*, with three homeologs each[30].

Homeolog-specific RNAseq analysis showed that all six wheat *Med15* homologs are expressed in the first leaf (Fig. 3b and Supplementary Fig. 11); therefore, all six Med15 proteins could participate in forming the multi-protein Mediator complex. This recent duplication may form the basis of neo-functionalization of *Med15* in Triticeae species. We performed branch-specific tests for variable levels of selective pressure for the *Med15*, *Med15a*, and *Med15b* clades (Supplementary Fig. 12). Purifying selection for non-Triticeae species was strongest ($\omega_0 = 0.24$), whereas both *Med15a* and *Med15b* experienced relaxed purifying selection ($\omega_a = 0.38$, $\omega_b = 0.37$; Supplementary Table 9). Although both $\omega_a$ and $\omega_b$ were below 1.0, indicating a lack of recurrent positive selection, the relaxation in purifying selection may impact a subset of *Med15* codons.

**Differential gene expression in wild-type and mutant lines.** The role of *Med15* in transcriptional regulation suggests that steady-state or pathogen responsive transcription might contribute to suppression of stem rust resistance. To identify putative steady-state transcriptional targets of *Med15b.D*, we performed RNAseq profiling on uninoculated leaves of CTH, NS1, and NS2. A total of 229 differentially expressed genes were identified using a false discovery rate of 5% (Fig. 4a and Supplementary Table 10). The physical distribution of differentially expressed genes was overlaid on the wheat genome, uncovering two regions located on chromosomes 1A and 1D that were associated with 31 downregulated genes and 169 upregulated genes, respectively (Fig. 4b). These regions are homeologous and share approximately half of the

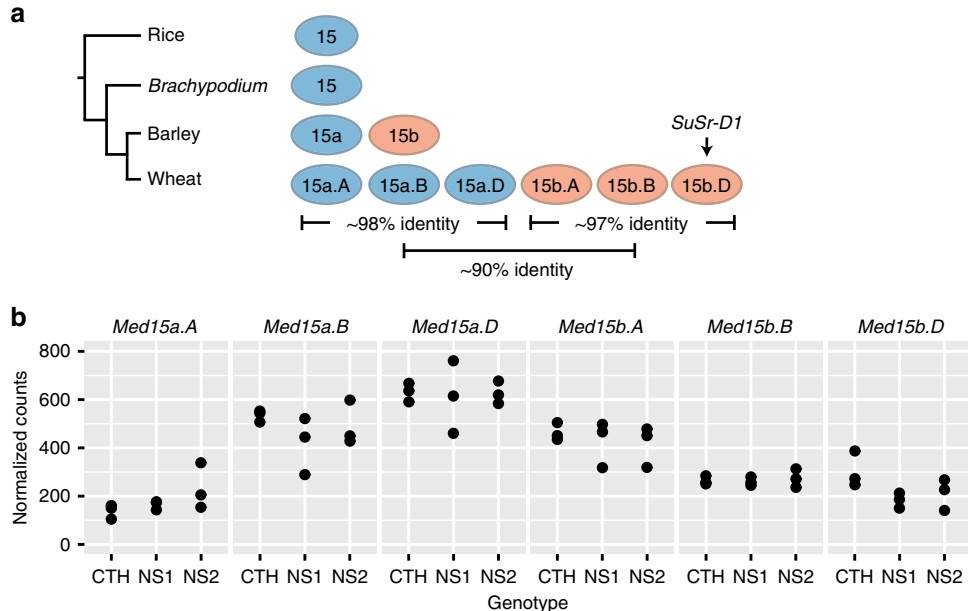

**Fig. 3 Expansion of the *Med15* gene family in the Triticeae. a** Rice and *Brachypodium distachyon* have a single copy of *Med15*, whereas barley and wheat have two and six copies of *Med15*, respectively. Rice contains an additional *Med15* gene (*Med15c*; LOC_Os04g03860) that is highly divergent in protein identity and not present in *B. distachyon*, barley, or wheat[31]. Homeologous *Med15a* and *Med15b* gene families exhibit 98 and 97% protein identity, whereas there was 90% protein identity between the families. **b** All six copies of *Med15* in wheat were expressed in leaf tissue. RNAseq profiling was performed in triplicate, with individual data points representing a single experiment.

gene content (242 genes) between these regions in the individual sub-genomes. Characterization of the physical intervals found that almost all genes within the region were segmentally co-regulated, with the chromosome 1A and 1D regions experiencing suppression and induction, respectively, whereas the home-ologous region on chromosome 1B was not modified (Fig. 4c, d and Supplementary Table 11).

To understand the impact on the transcriptional response in *Med15b.D* mutants relative to wild-type in the presence of *Pgt*, we performed RNAseq experiments on *Pgt*-inoculated and mock-inoculated seedlings of CTH, NS1, and NS2 with samples collected at 0 and 24 h post-inoculation (hpi). One hundred and thirty-nine, 19, and 56 genes were differentially expressed between *Pgt* and mock inoculated at 0 h post inoculation in seedlings of CTH, NS1, and NS2, respectively (Fig. 5a). After 24 h post inoculation, 8389 genes were differentially expressed in CTH, whereas NS2 had 2199 differentially expressed genes and NS1 had 55 using a false discovery rate of 5% (Fig. 5a, Supplementary Table 12). There was considerable overlap in differentially expressed genes in CTH versus NS2 (1435 genes) and NS1 (16 genes) (Fig. 5b). This suggests that differential gene expression at 24 hpi was not required for immunity to *Pgt* and that loss of function in Med15b.D severely impacted the transcriptional response of CTH to *Pgt*.

Many of the genes within the regions experiencing segmental co-regulation on chromosomes 1A and 1D in non-inoculated plants were also identified in both mock and *Pgt*-inoculated experiments. In the mock-inoculated experiment, comparison of CTH and NS1 found 188 of 284 (66%) and CTH and NS2 with 179 of 227 (79%) differentially expressed genes overlapping with non-inoculated differentially expressed genes on chromosomes 1A and 1D. At 24 hpi, CTH and NS1 had 10,035 differentially expressed genes after *Pgt*-inoculation, whereas CTH and NS2 had 668 differentially expressed genes. The high number of differentially expressed genes between CTH and NS1, and low number in CTH and NS2 reflected a lack of transcriptional response in NS1 after *Pgt*-inoculation. A total of 317 genes were

shared between these two comparisons, with 113 genes (36%) in the chromosome 1A and 1D regions.

The minor impact on steady-state gene expression levels in nonsense *med15* mutants in wheat parallels the lack of morphological differences across four of five growth parameters between the wild-type and mutants (Supplementary Fig. 13). In contrast, a null allele of *atmed15* (*nrb4-4*) in *Arabidopsis thaliana* exhibited pleiotropic morphological defects including chlorosis, delayed growth, and sterility[32]. Therefore, *Med15b.D* exhibits a homeolog-dominant suppression of immunity that is generally compensated by homeologs in its role as a co-activator of transcription. Given the apparent degree of homeologous compensation in bread wheat, *Med15* is a candidate target for gene editing as means to improve disease resistance with little morphological consequence.

**Discussion**

Disease resistance is a major goal of wheat breeding programs around the world. Keeping pace with pathogen evolution repre-sents a significant challenge. Close relatives of wheat are reser-voirs of genes that can be used for wheat improvement including disease resistance[22]. However, attempts to transfer disease resis-tance genes to hexaploid wheat are often unsuccessful because the resistance genes are suppressed[11–19]. Despite the many reports of suppressed resistance genes, there have been only a few reports describing the genetic basis of suppression. A molecular mechanism for suppression of disease resistance was previously described in wheat involving the suppression of powdery mildew resistance by direct interaction of NLR plant immune receptors encoded by alleles of *Pm3* or *Pm8* (a homeolog of *Pm3*)[33]. Interaction of homeologous genes was also implicated in the suppression of *Lr23*[15].

Suppression of stem rust resistance by *SuSr-D1* follows a dif-ferent mechanism rather than that observed in the *Pm3/Pm8* interactions. Med15b.D, encodes a subunit of the Mediator com-plex, which is a coactivator of transcription through its interaction with RNA polymerase II and general and specific transcription

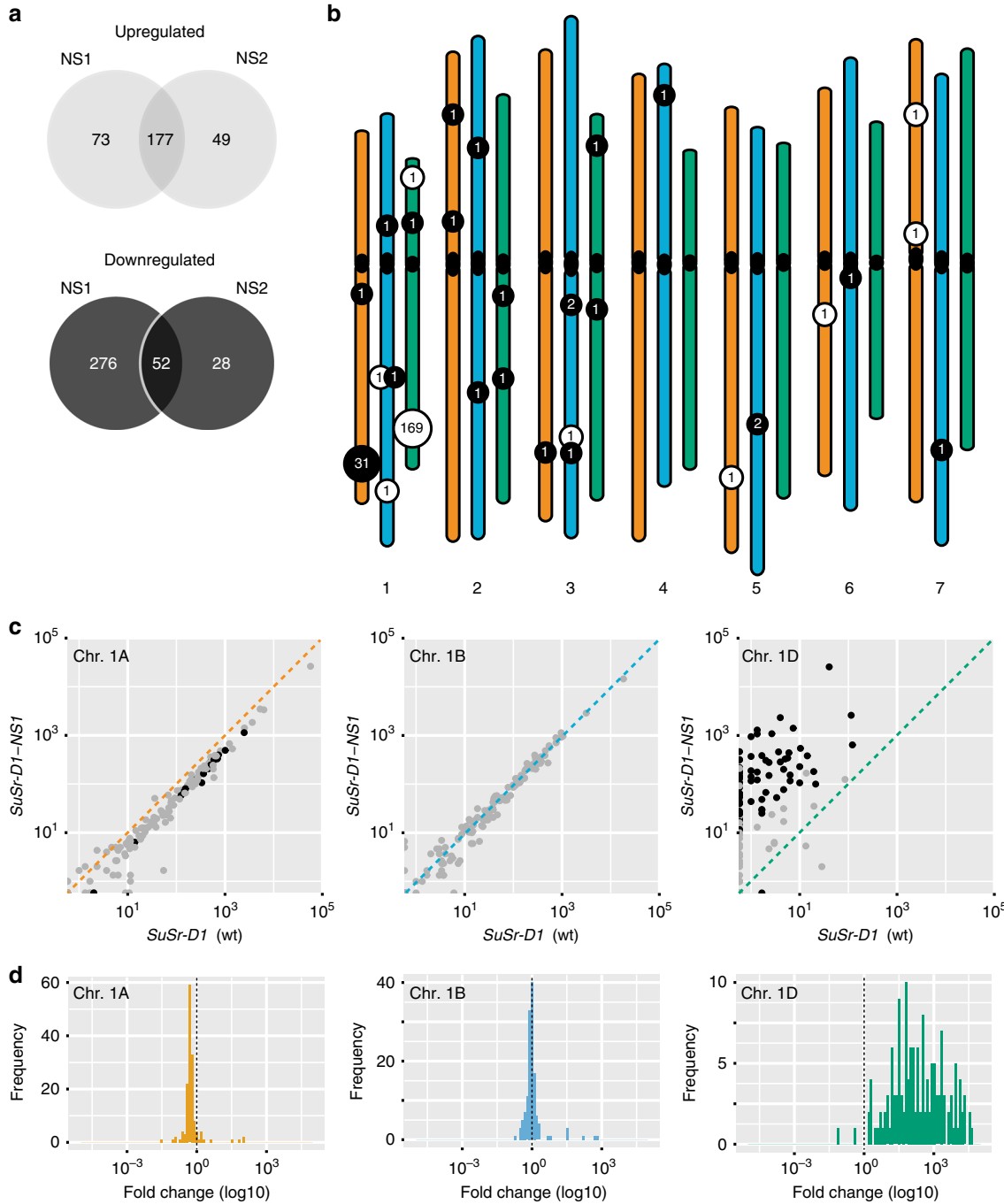

**Fig. 4 Mutation in *SuSr-D1* differentially regulates homeologous regions on chromosome group 1. a** Venn diagrams showing RNAseq-based differentially expressed (DE) genes between wild-type (CTH) and *SuSr-D1* mutants NS1 and NS2 using a false discovery rate (FDR) of 5%. Top, upregulated genes (higher expression in mutant relative to wild-type). Bottom, downregulated genes (lower expression in mutant relative to wild-type). **b** Physical distribution of DE genes across the chromosome of wheat. The majority (87%) of DE genes localize to homeologous regions on chromosomes 1A and 1D. Upregulated and downregulated genes are shown in white and black filled circles, with A, B, and D genomes shown in orange, blue, and green, respectively. Below, Triticeae chromosome groups. **c** Pair-wise plots of leaf expression levels for homeologous chromosome group 1 regions. Black circles represent differentially expressed genes (FDR 5%), whereas gray circles are not differentially expressed. Diagonal lines indicate equal expression levels. *x*-axis (wild-type, *SuSr-D1*) and *y*-axis (mutant, *SuSr-D1-NS1*) are $\log_{10}$-transformed based on transcripts per million. **d** Histograms of fold change in gene expression for all genes shown in Fig. 4c with non-zero expression in both wild-type and mutant NS1.

factors to coordinate transcription in eukaryotes[34,35]. The complex in Arabidopsis contains 33 subunits, and mutations within the gene affect a wide range of developmental processes as well as biotic stress responses[36,37]. However, there is no report of activated expression of race-specific disease resistance caused by mutations in Mediator subunits. In wheat, we identified six *Med15*

homologs encompassing two families, *Med15a* and *Med15b*, with three homeologs each. All copies of *Med15* were expressed in CTH but only *Med15b.D* suppressed stem rust resistance providing an example of homeolog-specific function in wheat.

At the interface of RNA polymerase II, general transcription factors, and specific transcription factors, the Mediator complex

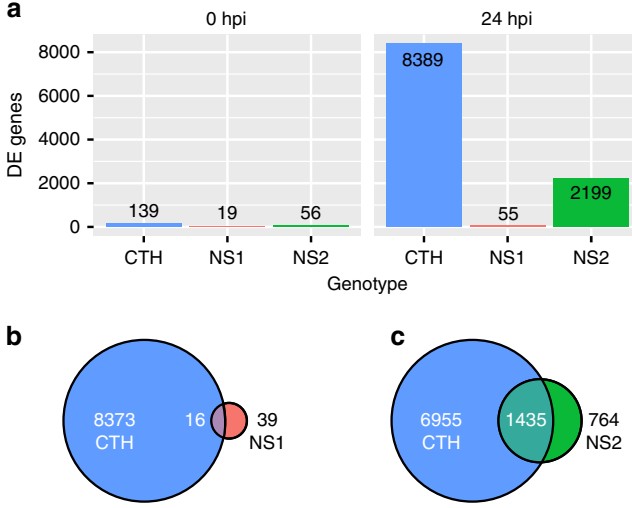

**Fig. 5 Mutation in *SuSr-D1* severely impacts the transcriptional response of CTH to *Pgt*. a** Number of differentially expressed genes (DE genes) between *Pgt*-inoculated and mock-inoculated wild-type *SuSr-D1* (CTH) and *SuSr-D1* mutants (NS1 and NS2) based on FDR of 5%. **b, c** Venn diagrams showing overlap of DE genes between wild-type and *SuSr-D1* mutants NS1 and NS2, respectively.

plays a central role in coordinating transcriptional regulator proteins and core promoters[38–40]. Most of our understanding of the Mediator complex in plant immunity has come from Arabidopsis[36,37], with subunits AtMED8, AtMED14, AtMED15-1, AtMED16, AtMED21, and AtMED25 having regulatory roles in plant immunity[41]. In addition, AtMED19a interacts with the nuclear-localized effector HaRxL44 from *Hyaloperonospora arabidopsidis* (an oomycete pathogen of *A. thaliana*), leading to degradation of MED19a in a proteasome-dependent manner[42]. High-throughput transcriptome experiments have been performed with several Mediator subunit mutants. Microarray-based transcriptome analysis on wild-type *AtMEDIATOR15-1* (AT1G15780; *NON-RECOGNITION-OF-BTH 4*; *NRB4*) and mutants *nrb4-2* (non-synonymous mutant) and *nrb4-4* (gene disruption by T-DNA) found substantial differential gene expression in *nrb4-4* (4416 genes), whereas *nrb4-2* had no differentially expressed genes relative to wild-type[32]. The *nrb4-4* mutation is a T-DNA insert in an intron immediately after the start codon, likely causing a significant impact on gene expression[32]. *nrb4-4* exhibits pleiotropic morphological defects including chlorosis, delayed growth, and sterility, and impaired salicylic acid and immune signaling[32]. Using time course microarray-based transcriptome analysis on wild-type and *atmed14-1* inoculated with an avirulent isolate of the bacterial pathogen *Pseudomonas syringae* pv *tomato* DC3000/avrRpt2, Zhang and colleagues found that *atmed14-1* responded more slowly at the transcriptional level to the presence of *P. syringae* pv *tomato* DC3000/avrRpt2[43]. The total number of differentially expressed genes was several thousand over the 0–12 hpi time-course[43], unlike the substantially perturbed/abolished expression observed in *med15b.D* mutants. The impact of gene knockouts in Mediator complex subunits on global transcriptional reprogramming remains unclear for other subunits, as several studies have only assessed reporter genes, such as *PATHOGENESIS-RELATED GENE1* (PR1)[44–46]. Similar to *atmed14*, *atmed15*, and *atmed16*, nonsense mutations in *med15b.D* led to reduced (NS2) or lost (NS1) transcriptional response to *Pgt*. Unlike *atmed14*, *atmed15*, and *atmed16*, the impaired/abolished transcriptional response at 24 hpi in *med15b.D* mutants was associated with immunity. While it remains unclear whether the transcriptional response is simply delayed or completely abolished

relative to wild-type at other time points, the lack of expression in *med15b.D-NS1* indicates that the transcriptional response at 24 hpi is not required for resistance to *Pgt*. All *med15b.D* mutants carried nonsense mutations in *Med15b* indicating that resistance (or suppression of transcriptome reprogramming) requires loss or truncation of the protein subunit.

The majority of differentially expressed genes in steady-state expression levels between wild-type and *med15b.D* mutant plants were associated with 25.7 Mb and 28.9 Mb intervals on chromosomes 1A and 1D, respectively. While genes in the chromosome 1A interval underwent a 50% reduction in expression, genes in the chromosome 1D interval showed substantial induction in expression ranging from 2 to >200 fold-change. Five NLR-encoding genes (the most prevalent cytoplasmic/nuclear immune receptors in plants) were located within the upregulated interval on chromosome 1D, and were potential candidate genes contributing to immunity to *Pgt*. This hypothesis will require additional genetic mapping to demonstrate that immunity to *Pgt* maps to the chromosome 1D interval.

The Mediator complex has a known role in fine-tuning gene-specific and pathway-specific transcriptional reprogramming[47]. In this study, we established a potential novel role in segmental coregulation, for which there is no known mechanism in eukaryotes that regulates genes over a >25 Mb interval. In mammals, spatial partitioning of the genome into topologically associating domains coordinates gene expression, but these regions range from 40 kb to 3 Mb[48]. The regulatory processes that contribute to this coregulation could involve modification of chromatin architecture such as the formation of chromatin loops or condensation of heterochromatin[49]. While well characterized in yeast and mammals, little is known about the role of the Mediator complex in controlling chromatin loops and heterochromatin in plants. Interestingly, the homeologous chromosome 1B interval (51 Mb) did not show any modification in gene expression, suggesting genomic features such as sequence or size could differentiate this region from the chromosome 1A and 1D intervals. Collectively, the identification of a suppressor of immunity in wheat demonstrates the complexity of interactions between its sub-genomes, and identifies a novel approach at improving agriculture through the removal of suppressors that negatively interact in wheat.

## Methods

**Plant materials.** The wheat accession Canthatch-K (CTH-K) is a selection of Canthatch (CTH) made by E. R. Kerber (Agriculture and Agri-Food Canada, retired). CTH-K was previously used in mutagenesis experiments to generate independent mutant lines, RL5863 (NS1) and RL5864 (NS2), that carry null mutations for *SuSr-D1* on chromosome arm 7DL[25]. Additional mutants from CTH designated W01, W02, W03, W06, W07, W10, W11, and W12 were also included in the study[26]. Crosses were made between Thatcher and NS1, and between Thatcher and NS2. Thatcher is the recurrent parent of CTH. The F₁ plants from each cross were used to make doubled haploid (DH) populations referred to as the NS1 DH and the NS2 DH populations, respectively, for the purpose of mapping *SuSr-D1*. DH populations were generated using a maize pollination method[50]. Little Club wheat was used as susceptible check for stem rust tests. Lines CTH-K, Thatcher, NS1, NS2, CTH ditelosomic 7DL (CTH-7DL), and Williams et al.[26]. CTH and mutants W01, W02, W06, W07, W10, and W11 were used for chromosome isolation and sequencing experiments (Supplementary Table 3). The cytogenetic stock, CTH ditelosomic 7DS (CTH-7DS), was used in seedling stem rust tests.

**Assessment of plant development.** Assessment of plant developmental parameters of CTH-K, NS1, and NS2 was performed through the growth of six to seven independent plants in three different greenhouses (S52, S55, S59) in Norwich, UK from 16 April 2018 to 1 August 2018. All plants were sown on the same date and randomly assigned positions within the greenhouse. Assayed growth parameters included complete emergence of the flag leaf (Zadoks scale 47), first emergence of awns (Zadok scale 49), one-quarter spike emergence (Zadoks scale 53), total number of seeds harvested, and 1000 grain weight (Fig. 5). For 1000 grain weight, the weight was based on three randomly sampled technical measurements using

100 seeds. ANOVA was performed in R (3.5.1) *aov* using the factors genotype, location, and their interaction. Tests of homogeneity of variance (Levene test) and normality (Shapiro-Wilk test) were performed on individual traits. Pairwise differences were assessed using adjusted *p* values using Tukey Honest Significant Differences. All traits passed these tests, except for 1000 grain weight. For 1000 grain weight, the non-parametric Kruskal-Wallis test was used, with pairwise differences determined using the Wilcoxon rank sum test, adjusting *p* values using the Benjamini-Hochberg approach[51].

**Seedling and field tests for stem rust resistance.** NS1, NS2, CTH-K, Thatcher, NS1 DH population, and NS2 DH population were tested for stem rust reaction at the seedling stage. Seedlings where inoculated after the first leaf was fully emerged using *Pgt* race QTH (C25) (AAFC-Morden isolate 1347). Classification of *Pgt* races is based on a previously described nomenclature system[52,53]. Urediniospores were suspended in light mineral oil (Bayol 55, Imperial Oil Canada, Toronto, Canada) and sprayed onto seedlings. Inoculated seedlings were incubated for 16 h in dew chambers at 100% relative humidity and then were dried slowly under light to promote infection. The seedlings were transferred to a greenhouse and grown at approximately 20 °C with 16 h of light daily including supplementary fluorescent lights. Seedlings were rated for infection type (IT) 14 days post-inoculation following the 0–4 scale previously described[10]. ITs 0–2 were classified as resistant and ITs 3–4 were classified as susceptible. Seedlings of CTH-K, CTH-7DL, CTH-7DS, NS1, and NS2 were tested with *Pgt* races QTHJC (C25; isolate 1541), TPLKC (C33; isolate 1427), TPMKC (C53; isolate 1373), QCCSC (C56; isolate 901), RTHJF (C57; isolate 1561), SPMMC (C74; isolate 972), RKQQC (C35; isolate 1312), MCCFC (C17; isolate 1541), QFCSC (isolate 06ND76C), QTHJC (isolate 75ND717C), TTKSK (isolate 04KEN156/04), TTKST (isolate 06KEN19v3), TTTSK (isolate 07KEN24-4), TRTTF (isolate 06YEM34-1), TPMKC, 21-2,5 (49) and 34-1,2,3,5,6,7 (313) as described above[52,54–56]. CTH-K, NS1, and NS2 were tested in field trials in 2009, 2010, 2011, 2012, 2016, and 2018 and CTH-W, W01 at Njoro, Kenya that were inoculated with Ug99 races of *Pgt* following previously described procedures[57]. Field plots were rated for stem rust severity and infection response using the modified Cobb scale[58].

**Chromosome isolation and DNA amplification.** Chromosome 7D from CTH, Thatcher, NS1, NS2, and five CTH mutants (W01, W02, W06, W07, and W10)[26] and chromosome arm 7DL from CTH-ditelosomic 7DL (CTH-DT7DL), were isolated by flow cytometric sorting (Supplementary Figs. 3, 4; Supplementary Table 3)[27]. Suspensions of intact mitotic metaphase chromosomes were prepared from synchronized root tips of young seedlings according to Vrána et al.[59]. Prior to chromosome analysis by flow cytometry, fluorescence in situ hybridization in suspension (FISHIS) was used to label GAA microsatellites by FITC following the protocol of Giorgi et al.[60], with modifications. Briefly, chromosomal DNA was denatured by adding 10 M NaOH to the solution to reach pH 12.8–13.3. Following incubation at room temperature for 15 min, pH of the solution was changed to 8.5–9.1 using 1 M Tris-HCl (pH 7.5) and the sample was incubated on ice for 1 min. Then, $(GAA)_7$-FITC probe was added to the suspension to final concentration 4.6 ng/μL and the sample was incubated in darkness at room temperature for 1 h. After FISHIS, chromosomal DNA was stained by fluorochrome DAPI (4′,6-diamidino-2-phenylindole) at 2 μg/mL final concentration and the suspension was analyzed by FACSAria SORP II flow sorter (BD Biosciences, San Jose, USA) at rates of 1000–2000 particles/s.

A total of 1000 particles from the population putatively representing 7D or 7DL were sorted onto a microscope slide and chromosome identity was checked under a fluorescence microscope based on the GAA labeling pattern. Once a target population was confirmed, three independent samples of 1000 chromosomes were sorted onto microscope slides to estimate the extent of contamination by other chromosomes using FISH with probes for Afa-family repeat and $(GAA)_7$ microsatellites (Supplementary Table 3)[61]. Three batches of 33,000 and 66,000 copies of chromosome 7D and the 7DL telosome, respectively, were then sorted from each sample into PCR tubes containing 40 μL sterile deionized water. The number of sorted chromosomes was determined so that an equivalent of approximately 50 ng DNA was obtained per sample. To produce DNA for sequencing, chromosomal DNA was purified and amplified by Illustra GenomiPhi V2 DNA Amplification Kit (GE Healthcare, Piscataway, USA) according to Šimková et al.[62]. Each sample of sorted chromosomes was amplified separately and three amplification products from the same chromosome were pooled to reduce possible amplification bias. DNA amounts thus obtained are listed in Supplementary Table 3.

**Chromosomal DNA sequencing and assembly.** PCR-free libraries for Illumina sequencing were prepared by Novogene (Hong Kong, China) using the NEBNext Ultra II DNA Library Prep Kit for Illumina (New England BioLabs, Ipswich, USA) following the manufacturer's protocols and index codes were added to attribute sequences to each sample. Sequencing was performed for chromosome 7D isolated from CTH-K, NS1, NS2, Williams et al.[26] wild-type (CTH-W) and mutants W01, W02, W06, W07, W10, Thatcher, and isolated chromosome arm 7DL from CTH-DT7DL. Samples were sequenced on an Illumina HiSeqX (Illumina, San Diego, USA) to generate 150 bp paired-end reads with two samples per lane except for

CTH-K and CTH-W with 250 bp paired-end reads which were each run singly in a lane (HiSeq2500). Data quality from paired end reads was assessed with FastQC and reads were removed using Trimmomatic (v0.36) with parameters set at ILLUMINACLIP:TruSeq3-PE.fa:2:30:10, LEADING:5, TRAILING:5, SLIDINGWINDOW:4:15, and MINLEN:150[63]. These parameters remove all reads with the adapter sequence, ambiguous bases, or a substantial reduction in read quality. The de novo assembly of chromosome 7DL was performed using Edena (v3.131028) with default parameters, use of paired end relationships, and a minimum overlap of 100 bp[64,65].

**Identification of genes and SNVs in the *SuSr-D1* interval.** The assembly of chromosome arm 7DL from CTH-DT7DL was masked for repetitive sequence using RepeatMasker (version open-4.0.5; repeat library version 20140131)[66]. A self-alignment of reads from wild-type CTH, NS1, and NS2 chromosome 7DL was performed using bwa aln with default parameters (version 0.7.10-r789). Paired end reads were merged using bwa sampe. Samtools (version 1.9) was used to select only paired end reads, the removal of duplicate sequences, and generate pileup files for SNV identification. The VarScan (version 2.3.8) pileup2snp command was used to identify polymorphisms relative to the reference sequence. Alignments for mutants NS1 and NS2 were performed using identical parameters[67]. Python scripts mutant_flow_sorting_analysis.py and analyze_HQ_SNPs.py were used to identify SNVs that differed between wild-type and mutants NS1 and NS2. In wild-type and mutant, a requirement of at least ten reads was used for genotype calls. CTH mutants NS1 and NS2 were generated using EMS mutagenesis. EMS induces alkylation of guanine (G), which results in conversion of G to A (adenine) due to pairing of $O^6$-ethylguanine with thymine (T). After replication, equal numbers of mutations should be observed as conversions of cytosine (C) to T. For an initial screen of SNVs, two parameter sets were selected: the strict set using parameters of 95% mutant and 5% wild-type allele frequency and the relaxed set using 70% mutant and 20% wild-type allele frequencies (Supplementary Table 5).

Tophat (version 2.0.9) was used to perform spliced alignment of RNAseq reads derived from CTH, NS1, and NS2[68]. Default parameters were used with the exception of maximum intron length of 20,000. In total, 27,664,945 reads were used. Approximately 4.0% of reads aligned to the chromosome 7D assembly. Aligned reads were used to generate gene models using cufflinks (version 2.1.1) with default parameters[69]. Transdecoder.LongOrfs (version 2.0.1) was used to identify open reading frames (ORFs) in protein encoding genes[70]. BLASTn (version 2.2.26) was used to align contigs onto the reference wheat genome (IWGSC RefSeq v1.0) using default parameters with the exception of no filtering of the query sequence. The Python script link_position_expression.py was generated to export all ORFs from gene models, independently integrating SNVs in NS1 and NS2. ORFs were translated and mutations that generated non-synonymous changes were identified. All mutations shared between NS1 and NS2 relative to the reference were removed, under the assumption they were due to contamination or were shared mutations relative to CTH-K. Selection of SNVs for analysis on NS1 and NS2 DH populations was prioritized based on contigs with SNVs in leaf expressed genes that led to non-synonymous changes that were evenly spread across chromosome arm 7DL. The complete analytical pipeline and custom scripts are available on a GitHub repository (https://github.com/matthewmoscou/Canthatch).

**Genetic mapping of *SuSr-D1*.** The populations segregating for wildtype and mutant *SuSr-D1* were fixed for the suppressed *Sr* genes. *SuSr-D1* mutant lines[25] NS1 and NS2 were crossed with Thatcher, the recurrent parent of CTH, and the hybrids were used to generate doubled-haploid (DH) populations. The Thatcher × NS1 (*N* = 159) and Thatcher × NS2 (*N* = 129) DH populations segregated for reaction to stem rust race QTHJC, and both populations fitted 1:1 single gene ratios (*p* = 0.48 and *p* = 0.06, respectively). A total of 35 Kompetitive Allele Specific PCR (KASP) markers (LGC, Teddington, UK) based on EMS-induced SNVs[28] were assayed on the two DH populations to localize *SuSr-D1* to a 1.3 cM genetic interval flanked by markers kwh239 and kwh281 (Fig. 2a, b and Supplementary Tables 5, 6)[29]. Linkage maps of chromosome arm 7DL were constructed for both DH populations using MapDisto[71]. Genetic distances were calculated using the Kosambi mapping function[72]. The physical positions of the SNV markers were determined by locating flanking sequences on the wheat reference genome assembly (IWGSC RefSeq v1.0) using BLAST. To compare the map positions of the cross-specific SNV markers and *SuSr-D1* with commonly used markers, markers from the wheat 90k iSelect consensus map[73] on chromosome arm 7DL were also located on the IWGSC RefSeq v1.0. Flanking markers were anchored on the International Wheat Genome Sequencing Consortium reference sequence of Chinese Spring (IWGSC RefSeq v.1.0) for chromosome 7D[30]. A minimum physical interval of 1079 kb in Chinese Spring wheat corresponded to the region in CTH carrying *SuSr-D1*[30] (Fig. 2c, Supplementary Table 7). De novo assembled contigs from CTH, NS1, and NS2 were aligned to the *SuSr-D1* physical region. Chinese Spring and CTH share an isogenic haplotype for the interval, facilitating the identification of SNVs based on the Chinese Spring reference annotation. The region carrying *SuSr-D1* contained 17 high confidence and 10 low confidence genes (Fig. 2c and Supplementary Table 4). Only one gene carried mutations in both NS1 and NS2 generating nonsense mutations in *Med15b.D*, a member of the Mediator complex (Fig. 2d)[37]. SNVs responsible for nonsense mutations in *Med15b.D* cosegregated with the phenotype

in both DH populations (Fig. 2b). De novo assembly of flow-sorted chromosomes from eight mutants identified two mutants with large chromosome deletions and six mutants with nonsense mutations in *Med15.D* (Fig. 2d and Supplementary Figs. 7, 8)[74]. The nonsense mutations in the encoded protein ranged from changes at amino acid positions 33 (W06) to 820 (W10 and W11) (Supplementary Figs. 7, 9). A common mutation in W10 and W11 suggested they were sib lines from the mutagenesis experiment[26] as both lines possessed common SNVs across many contigs. Taken together, independent mutations confirmed that inactivation or deletion of *Med15b.D* was responsible for the activation of suppressed stem rust resistance.

**Identification of SNVs in additional CTH mutants.** De novo assemblies for each of the wild-type accessions, CTH-W and CTH-K, were created from their quality-controlled, paired-end sequencing reads using CLC Genomics Workbench (v9) with a length fraction of 0.95, a similarity fraction of 0.98 and the remaining parameters as default. Assembly FASTA files consisted of 767,884 contigs spanning approximately 841 Mbp and 749,802 contigs spanning approximately 806 Mbp for CTH-W and CTH-K, respectively. Assemblies were masked for repeat sequences using RepeatMasker (v4.0.6)[66] and the custom Triticeae, non-redundant repeat library trep-db_nr_Rel-16.fasta (release 16) downloaded from the TRansposable Elements Platform (TREP) (http://botserv2.uzh.ch/kelldata/trep-db/index.html). Alignment to wild-type assemblies of quality-controlled reads from wild-types CTH-K and CTH-W, and mutants W01, W02, W06, W07, W10, was performed using bwa aln followed by bwa sampe to map paired-end reads using default parameters (bwa v0.7.15). Samtools (v0.1.19) was used for subsequent processing of alignments to select only paired-end reads, remove duplicates and generate pileup files for SNV calling. SNV identification was performed using an in-house pipeline. The program Noisefinder.pyc was used on alignments for the wild-types (CTH-W, CTH-K) to detect regions with high density noise possibly due to misalignments and presence of allelic variants. SNVs were logged in all wild-types and mutants using the program SNPlogger.pyc, with noisy regions detected by Noisefinder.pyc being masked. Subsequent discovery of candidate sequences was performed using SNPtracker.pyc to shortlist sequences containing polymorphisms coinciding across multiple mutants. A threshold of 80% mutant and 20% wild-type allele frequency was used and variant positions detected in the wild-types were masked from the analysis. In both wild-type assemblies, a contig of approximately 16 kb was identified for which all five mutants presented independent SNVs that were canonical for EMS mutagenesis (C->T/G->A) (Fig. 2 and Supplementary Fig. 7). Custom programs used in this analysis are available on GitHub (https://github.com/TC-Hewitt/MuTrigo).

**Protein sequence and domain analyses.** Amino acid frequency was assessed using a sliding window of 50 amino acids to identify regions that were rich for specific amino acids using QKutilities_protein_analysis.py (https://github.com/matthewmoscou/QKutilities). Motif and structure-based domain analysis was performed using InterProScan[75,76] and Phyre2[77]. Coiled coils predictions were made using Coils[78], Marcoil[79], and Pcoils[80].

**Phylogenetic and molecular evolution analyses.** Multiple data sources were used to identify homologs of *Med15*. Rice (*O. sativa*) and *B. distachyon Med15* homologs were identified using the Department of Energy-Joint Genome Institute Phytozome database (https://phytozome.jgi.doe.gov), barley (*H. vulgare*) homologs from the 2017 genome sequence[81], and bread wheat (*T. aestivum*) from the reference genome sequence (IWGSC RefSeq v1.0). We accessed publicly available leaf RNAseq data for eight grass species (Supplementary Table 13)[82,83], performed de novo transcriptome assembly using Trinity[84] (version r20140717) using the parameters "--min_kmer_cov 2 --normalize_max_read_cov 20 –trimmomatic", and identified *Med15* homologs using *BLAST*. PRANK (v.140603) and RAxML (v8.2.9) were used for codon-based alignment and phylogenetic tree construction, respectively[85–87]. Default parameters were used for PRANK and RAxML parameters include the GTRCAT model using rice *Med15* as an outgroup. A total of 2000 bootstraps were performed. In addition, pairwise comparisons were made between DNA and protein alignments (Supplementary Table 14). Molecular evolutionary analyses were performed with PAML (v4.8)[88]. A reduced phylogenetic tree based on a requirement of 90% coverage was used for estimating $\omega$ ($d_N/d_S$), as several sequences lacked sufficient coverage of *Med15* due to truncated transcript assemblies (Supplementary Fig. 12).

**RNAseq.** Two RNAseq experiments were performed. The first experiment included non-inoculated seedlings of CTH, NS1, and NS2, and the second experiment included mock-inoculated and *Pgt*-inoculated seedlings of CTH, NS1, and NS2. For the non-inoculated seedlings, first and second leaf tissue was harvested at 10 days after sowing of CTH, NS1, and NS2 grown in the greenhouse. For the second experiment, mock (mineral oil application) and *Pgt* isolate QTH inoculation was performed after the first leaf was fully elongated. Tissue was collected from CTH, NS1, and NS2 at 0 hpi and 24 hpi. Both experiments were performed using three independent biological replicates. Tissue was flash frozen in liquid nitrogen and stored at −80 °C. Tissues were homogenized into fine powder in liquid nitrogen-chilled pestle and mortars. RNA was extracted, purified, and assessed for quality as

described previously[89]. RNA libraries were constructed using TruSeq RNA Library Prep Kit v2 (Illumina). Barcoded libraries were sequenced using a 150 bp paired-end reads. All library preparation and sequencing was performed at Novogene. Initial quality was assessed using FastQC and reads were trimmed using Trimmomatic (v0.36) with parameters ILLUMINACLIP:TruSeq3-PE.fa:2:30:10, LEADING:5, TRAILING:5, SLIDINGWINDOW:4:15, and MINLEN:36. RNAseq data quality was assessed with FastQC. Read mapping for expression analysis was performed using Kallisto (v0.43.1) and BBmap (v37.77). Default parameters were used for kallisto, whereas subgenome-specific read mapping with BBmap was carried out using parameters set to require 100% identity (perfectmode = t), unambiguous mapping (ambiguous = toss), and proper paired reads only (samtools view –f 2). Differential expression (DE) analysis was performed DEseq2 (Bioconductor v3.2) using default parameters[90]. Multiple hypothesis testing was controlled using a false discovery rate of 5%. To identify putative homeologs in wheat, clustering of high and low confidence proteins from the IWGSC RefSeq 1.0 was performed using CD-HIT, with a requirement of 90% or greater identity[30].

**Arabidopsis microarray reanalysis.** Data from Canet et al.[32] was obtained from EBI ArrayExpress experiment E-MEXP-3602. This data set includes transcriptome analysis of *A. thaliana* wild-type (Col-0) and a nonsynonymous EMS mutant *nrb4-2* (3-week-old plants), and T-DNA mutant *nrb4-4* (5-week-old plants) in replicated experiments (three replicates). Raw CEL files were normalized using R/BioConductor *rma* package[91]. Pairwise comparisons using log₂ transformed expression data between wild-type and mutants were performed using R *t*-tests with multiple hypothesis testing controlled using *q*-value[92] at false discovery rate of 20%.

**Reporting summary.** Further information on research design is available in the Nature Research Reporting Summary linked to this article.

## Data availability

All high-throughput sequencing data described in this manuscript have been deposited in the NCBI BioProject "PRJNA401266 [https://www.ncbi.nlm.nih.gov/bioproject/PRJNA401266]" and European Nucleotide Archive (ENA) "PRJEB23265 [https://www.ncbi.nlm.nih.gov/bioproject/?term=PRJEB23265]". Illumina sequencing data for flow-sorted chromosomes are deposited in sequential NCBI SRR accessions SRR6001708 to SRR6001719 and ENA ERS1988338 to ERS1988347, de novo assemblies of flow-sorted chromosomes for CTH-K (NCBI "NTGG00000000 [https://www.ncbi.nlm.nih.gov/nuccore/NTGG00000000.1/]", ENA "ERS1988348 [https://www.ebi.ac.uk/ena/data/view/GCA_900235935.1]"), CTH-W (ENA "ERS1988349 [https://www.ebi.ac.uk/ena/data/view/GCA_900235945.1]"), NS1 (NCBI "NTGH00000000 [https://www.ncbi.nlm.nih.gov/nuccore/NTGH00000000.1/]"), and NS2 (NCBI "NTGI00000000 [https://www.ncbi.nlm.nih.gov/nuccore/NTGI00000000.1/]"). RNAseq sequencing data are deposited in NCBI SRR accessions SRR6003621 to SRR6003628 and SRR10426855 to SRR10426890. The source data underlying Figs. 1, 2b, 3b, 4, and 5, and Supplementary Figs. 1, 4, 7, 10–13, and Supplementary Table 9 have been deposited in a "Figshare project [https://figshare.com/projects/Stem_rust_resistance_in_wheat_is_suppressed_by_a_subunit_of_the_Mediator_complex/28056]".

## Code availability

Bioinformatic analyses and scripts can be found on Github https://github.com/matthewmoscou/Canthatch, https://github.com/matthewmoscou/QKutilities, and https://github.com/TC-Hewitt/MuTrigo.

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

## Acknowledgements

This work was supported in part by funds provided through Agriculture and Agri-Food Canada, Western Grains Research Foundation, Biotechnology and Biological Sciences Research Council (BB/P012574/1, BB/P016855/1), National Science Foundation (NSF BREAD program IOS-0965429), CSIRO, Gatsby Foundation, and the ERDF project "Plants as a tool for sustainable global development" (No. CZ.02.1.01/0.0/0.0/16_019/0000827). We thank Peter Dodds and Jeff Ellis for critical review of the manuscript prior to submission. We acknowledge the technical expertize of M. Popovic, G. Mardli, D. Edwards, T. Malasiuk, T. Zegeye, Z. Dubská, J. Weiserová, R. Šperková, W. Schnippenkoetter, and S. Chandramohan. We thank D. Klindworth and S. Xu for providing seed of mutant lines developed at the USDA (Fargo, North Dakota).

## Author contributions

Conceived and designed the experiments: C.H., M.M., W.S., J.D., L.H., B.W. and E.L. Performed transcript profiling: I.H.P. and N.U. Performed disease phenotyping: C.H., T.F., V.P., P.Z., M.R., Y.J., R.M., S.B., E.L. Performed chromosome flow sorting: J.V., M.K., J.D. Performed bioinformatic analyses: M.M., T.H., B.S. Designed software used in analysis: M.M., T.H., and B.S. Analyzed the data: C.H., M.M., W.S., T.H., B.S., V.P., P.Z., P.G., J.Z., R.M., S.B. and E.L. Developed mapping populations: C.H., V.P., and E.L. Contributed reagents/materials/analysis tools: P.Z. and R.M. Wrote the paper: C.H., M.M., W.S., J.D., T.H. and E.L.

## Competing interests

The authors declare no competing interests.
