## [Peer Review File · Nature Communications]

Editorial Note: This manuscript has been previously reviewed at another journal that is not operating a transparent peer review scheme. This document only contains reviewer comments and rebuttal letters for versions considered at Nature Communications .

REVIEWERS' COMMENTS:

Reviewer #2 (Remarks to the Author):

I have carefully read the revised manuscript and the rebuttal letter by the authors. The additional RNA-Seq experiments and data shown using Pgt-challenged and mock-inoculated wheat leaves have significantly improved this manuscript (Fig. 4 and 5). The authors now provide solid evidence that transcriptional reprogramming, at least at 24 h after pathogen inoculation, is not required for immunity to stem rust. This new result is important and will be of wider interest. Consequently, the original claim(s) of this study have been significantly toned down and the much improved Discussion with a nuanced data interpretation in the wider context of known plant Mediator activities now is appealing to a broad readership in the life sciences. All my reservations of the earlier manuscript version have been resolved.

Although future work is needed to understand how loss of wheat Med15b.D activates stem rust resistance and whether this activation is directly or indirectly linked to the observed segmental coregulation on chromosome 1A and 1D regions, I enthusiastically recommend publication of this revised manuscript.

REVIEWERS' COMMENTS:

Reviewer #2 (Remarks to the Author):

I have carefully read the revised manuscript and the rebuttal letter by the authors. The additional RNA-Seq experiments and data shown using Pgt-challenged and mock-inoculated wheat leaves have significantly improved this manuscript (Fig. 4 and 5). The authors now provide solid evidence that transcriptional reprogramming, at least at 24 h after pathogen inoculation, is not required for immunity to stem rust. This new result is important and will be of wider interest. Consequently, the original claim(s) of this study have been significantly toned down and the much improved Discussion with a nuanced data interpretation in the wider context of known plant Mediator activities now is appealing to a broad readership in the life sciences. All my reservations of the earlier manuscript version have been resolved.

Although future work is needed to understand how loss of wheat Med15b.D activates stem rust resistance and whether this activation is directly or indirectly linked to the observed segmental coregulation on chromosome 1A and 1D regions, I enthusiastically recommend publication of this revised manuscript.

RESPONSE: The authors thank the reviewer for the time invested in our manuscript during our initial submission to Nature Plants and our present version here in Nature Communications. Your suggestions have led to a stronger manuscript and we appreciate both your criticism and support.